# Therapeutic Strategies for RB1-Deficient Cancers: Intersecting Gene Regulation and Targeted Therapy

**DOI:** 10.3390/cancers16081558

**Published:** 2024-04-19

**Authors:** Mo-Fan Huang, Yuan-Xin Wang, Yu-Ting Chou, Dung-Fang Lee

**Affiliations:** 1Department of Integrative Biology and Pharmacology, McGovern Medical School, University of Texas Health Science Center, Houston, TX 77030, USA; mo-fan.huang@uth.tmc.edu (M.-F.H.); hatty110080901@gapp.nthu.edu.tw (Y.-X.W.); 2The University of Texas MD Anderson Cancer Center UTHealth Houston Graduate School of Biomedical Sciences, Houston, TX 77030, USA; 3Institute of Biotechnology, National Tsing Hua University, Hsinchu 300044, Taiwan; ytchou@life.nthu.edu.tw; 4Center for Precision Health, McWilliams School of Biomedical Informatics, The University of Texas Health Science Center at Houston, Houston, TX 77030, USA; 5Center for Stem Cell and Regenerative Medicine, The Brown Foundation Institute of Molecular Medicine for the Prevention of Human Diseases, The University of Texas Health Science Center at Houston, Houston, TX 77030, USA

**Keywords:** retinoblastoma, RB1, E2F, targeted therapy, therapeutic vulnerabilities, spliceosome, cell cycle, epigenetic regulators, ubiquitin-proteasome pathway

## Abstract

**Simple Summary:**

Addressing RB1-deficient cancers poses substantial challenges. Dysfunctional or deleted RB1 can enhance the proliferation and spread of different cancer types. Scientists strive to grasp the intricate role of RB1 in regulating various biological processes and molecular pathways to devise innovative therapeutic strategies tailored to RB1-deficient cancers. Recent advancements have revealed encouraging tactics for combating these malignancies, heralding a path toward more precise and efficacious treatments. This review underscores the pivotal role of RB1 in cancer research and highlights its potential as a focal point for personalized therapies.

**Abstract:**

The retinoblastoma (RB) transcriptional corepressor 1 (RB1) is a critical tumor suppressor gene, governing diverse cellular processes implicated in cancer biology. Dysregulation or deletion in RB1 contributes to the development and progression of various cancers, making it a prime target for therapeutic intervention. RB1′s canonical function in cell cycle control and DNA repair mechanisms underscores its significance in restraining aberrant cell growth and maintaining genomic stability. Understanding the complex interplay between RB1 and cellular pathways is beneficial to fully elucidate its tumor-suppressive role across different cancer types and for therapeutic development. As a result, investigating vulnerabilities arising from RB1 deletion-associated mechanisms offers promising avenues for targeted therapy. Recently, several findings highlighted multiple methods as a promising strategy for combating tumor growth driven by RB1 loss, offering potential clinical benefits in various cancer types. This review summarizes the multifaceted role of RB1 in cancer biology and its implications for targeted therapy.

## 1. Introduction

The retinoblastoma (RB) transcriptional corepressor 1 (RB1) stands as the first human tumor suppressor gene described by Dr. Alfred G. Knudson, who reported his observations on 48 cases of retinoblastoma, a rare eye cancer primarily affecting children, and associated reports [1]. Through the study of the RB1 tumor suppressor, Dr. Knudson suggested that individuals inherit one defective copy of a tumor suppressor gene (the “first hit”) and acquire a second mutation in the other copy of the gene (the “second hit”) over time, leading to the development of cancer. His “two-hit” hypothesis, aimed at elucidating the genetic basis of retinoblastoma, stands as a fundamental cornerstone to understand cancer genetics.

Since the discovery of RB1 decades ago, numerous studies on retinoblastoma and related non-ocular tumors have shed light on the molecular and genetic role of RB1 in cancer development and inheritance. These findings have significantly advanced our understanding of the pivotal role RB1 plays in cancer development. For instance, RB1 serves as a crucial regulator of cell cycle progression, acting as a safeguard against uncontrolled cell proliferation and tumorigenesis. Mechanistically, RB1 functions as a transcriptional repressor, fine-tuning the cell cycle by binding to and negatively regulating the function of E2F1/2/3 transcription factors (for a review, see references [2,3,4,5]). In addition to its role in modulating the cell cycle, RB1 has been discovered to play crucial roles in various processes unrelated to cell cycle control. The intricate interactions between RB1 and cellular pathways emphasize its integral role in maintaining genomic stability and curbing aberrant cell growth, positioning it as a prime target for cancer research and therapeutic interventions. Its importance is further underscored by its involvement in various cancer types, where mutations or dysregulation of RB1 contribute to the initiation and progression of diverse cancers [6].

The RB gene family comprises three key members: RB1 (p105), RBL1 (p107), and RBL2 (p130). While RBL1 and RBL2 are infrequently mutated in human cancers, RB1 mutations are prevalent across various cancer types, such as retinoblastoma, osteosarcoma, pinealoma, and melanoma [6]. These mutations contribute significantly to the oncogenic characteristics associated with RB1 loss of function. Research indicates that irreversible impairments in the functions of the RB1 tumor suppressor frequently predict poor prognoses in cancer patients. However, RB1 loss can also confer advantages, manifesting in diverse mechanisms and vulnerabilities during cancer development. These encompass the upregulation of RB1 targets initially functionally inactivated due to RB1 binding, the presence of genes whose inactivation leads to synthetic lethality with RB1 loss, or the simultaneous loss of neighboring genes resulting in concomitant synthetic lethality [7,8,9].

In this review, we will summarize the current evidence elucidating the pivotal role of RB1 in orchestrating both biological and pathological mechanisms across diverse cancer types, which not only enhances our understanding of the intricate molecular landscape due to RB1 loss but also lays the groundwork for the rational design of targeted therapeutic drugs. Specifically, we place a particular emphasis on potential strategies for targeted therapy that exhibit superior efficacy in the treatment of RB1-deficient cancers, offering a focused exploration of promising avenues in the pursuit of precision medicine.

## 2. Canonical Function of RB1 Tumor Suppressor

The RB1 tumor suppressor gene is initially characterized based on the germline predisposition of the pediatric eye tumor. Prior research has shown that RB1 functions as a key regulator in cell cycle progression [2]. Functional characterization of the RB1 tumor suppressor gene was originally observed to be able to arrest cells in the G1 phase of the cell cycle by inhibiting E2F1/2/3 transcription factor family activity. This suppression is abolished through the phosphorylation of RB1 facilitated by Cyclin C/CDK3, Cyclin D/CDK4/6, or Cyclin E/CDK2 [10] (Figure 1A). There is a growing body of literature that investigates many cellular roles of RB1 besides serving as a G1 checkpoint, including a contribution of cell fate determination [11], chromatin remodeling [12], apoptotic regulation [13,14], cellular differentiation control [4,15,16], DNA damage response [17], cellular senescence [18], homologous recombination [19], silencing of repetitive regions [20], centromere structure [21,22], pericentromeric structure and telomere maintenance [23,24,25,26], and immune responsiveness and evasion [27,28] (Figure 1B). Recent studies have further demonstrated RB1′s necessity in DNA double-strand break repair via canonical non-homologous end-joining (cNHEJ), wherein RB1 interacts with NHEJ components XRCC5 and XRCC6 [29], suggesting RB1 loss as a potential driver of structural genomic instability, contributing to cancer somatic heterogeneity and evolution. Furthermore, RB1 can interact with non-E2F transcription factors such as CEBPD [30], PU.1 [31], and androgen receptor (AR) [32], serving as a co-factor in modulating numerous gene expressions.

The developmental significance of RB1 function has been demonstrated, as indicated by the embryonic lethality observed in mice and the manifestation of defects in neurogenesis and hematopoiesis when the mice carry RB1 defective mutations [33,34]. In the context of lineage differentiation, RB1 plays a crucial role in determining cell fate, particularly by directing mesenchymal stem cells (MSCs) toward the osteoblast lineage. It achieves this by directly binding to Runx2 to activate its transcriptional function, thereby promoting osteogenic differentiation [35]. Deletion of p53 and RB1 in undifferentiated mesenchymal cells results in osteosarcomas expressing markers of multiple lineages. The range of tumors observed appears to vary depending on whether only p53 is lost or both p53 and RB1 are lost, with the latter resulting in a broader variety of tumors, including osteosarcomas, hibernomas, and sarcomas. The lack of RB1 promotes the differentiation of mesenchymal precursors into preadipocytes, which subsequently mature into brown adipocytes [35]. In line with in vivo animal studies, hereditary retinoblastoma iPSC-derived MSCs show impaired osteogenic differentiation compared to healthy iPSC-derived MSCs [36]. Moreover, these studies imply not only that the loss of RB1 may divert mesenchymal stem cells away from the osteoblastic lineage or induce their dedifferentiation but also suggest that RB1 holds therapeutic significance for bone and metabolic disorders. Further studies suggested the role of RB1 in cell gap junctional intercellular communication and osteoblast cell fate. These studies show that RB1 controls the expression of a diverse set of genes involved in cell adhesion and oversees the formation of adherens junctions necessary for cell adhesion. In certain types of tumors, inactivation of RB1 leads to both a disruption of cell cycle regulation, which facilitates initial tumor growth, and a disruption of cell-to-cell contacts [37,38]. Together, these results strongly support the role of RB1 in regulating cell differentiation and modulating normal developmental processes.

Furthermore, RB1 can interact with FOXM1 to modulate cellular differentiation and tumorigenesis. FOXM1 is a multifaceted transcription factor that coordinates various cellular processes crucial for tumorigenesis, such as cell cycle progression, antioxidant response, proliferation, invasion, metastasis, cellular senescence, stemness maintenance, and drug resistance [39,40]. Importantly, FOXM1 demonstrates both transcriptional activation and repression functions, the latter of which involves its interaction with RB1. In the G1 phase, FoxM1 forms a repressor complex with underphosphorylated RB1, recruiting DNMT3B to methylate the promoter regions of target genes, resulting in the suppression of genes associated with cellular differentiation, such as GATA3 and FoxA2, thereby inhibiting mammary luminal progenitor and hepatocyte differentiation, respectively [41,42,43]. In breast tumor cells, high expression of FoxM1 leads to the formation of a FOXM1/RB1 repression complex, which suppresses GATA3 and PTEN [44]. This regulation hampers differentiation and facilitates the adaptation of undifferentiated cells, fostering a pro-metastatic environment and ultimately leading to subsequent metastasis. Additionally, the FOXM1/RB1/DNMT3B transcriptional regulatory complex can suppress the expression of FOXO1, resulting in decreased levels of FOXO1 target p16, p21, and p27 in hepatocellular carcinoma (HCC) [45]. This subsequently leads to CDK-mediated RB1 phosphorylation and inactivation, overriding RB1-mediated senescence, and promoting the aggressive progression of HCC.

While many cellular functions and binding proteins have been generally revealed, the precise pathological roles of RB1 loss and its downstream effectors in distinct steps of tumorigenesis and various types of cancers largely remain unknown. It is necessary to consider the varied functions of RB1 across different cell types. While challenges such as lead optimization, pre-clinical testing, and clinical trials need to be addressed, historically, biomarker-matched drugs have exhibited high FDA approval rates. Therefore, it holds promise to target RB1/E2F downstream effectors in the treatment of RB1 mutation cancers.

## 3. Exploiting Vulnerabilities Stemming from RB1 Deficiency-Associated Mechanisms for Targeted Therapy

Based on the previous findings, four primary mechanisms associated with RB1 deficiency have emerged as promising candidates for targeted therapy development (Figure 2).

### 3.1. RB1-Deficient Cancers Present a Vulnerability in Spliceosomal Mechanisms

Alternative RNA splicing is a crucial process for eukaryotes [46], generating diverse RNA variants and protein isoforms from a single gene. Cancer cells often leverage the splicing machinery to create varied transcriptomes and unique splicing patterns [47,48]. Dysregulation of splicing factors, frequently upregulated in cancer cells, underscores the pivotal role of alternative splicing in cancer biology. Initially, RB1 was implied to be involved in spliceosome function, as indicated by the identification of the RB1/pp32 complex associating with components of the splicing machinery [49,50].

Tu J. et al. generated induced pluripotent stem cells (iPSCs; [51]) derived from patients with hereditary retinoblastoma and utilized this platform to reveal that the RNA processing pathway is induced under conditions of oncogene stress in RB1-mutant cells [36]. Through the analysis of transcriptomic and chromatin immunoprecipitation sequencing (ChIP-seq) data, they discovered that both RB1 and E2F3a co-regulate over a third of spliceosomal genes by binding to their promoters or enhancers. The clinical relevance of the RB1/E2F3a onco-spliceosome signature (REOSS) was demonstrated across various cancer types, with elevated expression observed in tumors exhibiting low RB1 and high E2F3a expression levels, which also correlated with poor prognosis in osteosarcoma (OS) patients. Significantly, pharmacological inhibition of the spliceosome using pladienolide B and SD6 in RB1-mutant cells led to widespread intron retention, reduced cellular proliferation, and compromised tumorigenesis ability, underscoring the therapeutic potential of targeting splicing dysregulation in RB1-deficient cancers. Notably, in support of the findings from Tu J. et al., the synthetic lethality screening performed by Oser MG et al. also identified that the depletion of RNA splicing factors, such as SNRPG, SNRPF, and U2AF1, leads to synthetic lethality in RB1-deficient small cell lung cancers (SCLCs) [52]. Importantly, targeting the RNA spliceosome is considered a potential strategy for cancer treatment, as evidenced by the proposed use of spliceosome inhibitors such as H3B-8800 in patients with myelodysplastic syndromes (MDS), acute myeloid leukemia (AML), or chronic myelomonocytic leukemia (CMML) with S3B1 mutation (clinical trial: NCT02841540) [53,54].

In summary, the dysregulation of alternative RNA splicing, particularly in RB1-deficient cancers, emphasizes the therapeutic potential of targeting the splicing machinery. The clinical relevance of the RB1/E2F3a onco-spliceosome signature underscores its prognostic significance across various cancer types.

### 3.2. Aurora Kinase Inhibitor-Induced Synthetic Lethality in RB1-Deficient Cancers

Targeting loss-of-function mutations or deletions in tumor suppressor genes is generally recognized as challenging. However, exploiting “gene-drug” synthetic lethality offers a promising opportunity for cancer therapeutic applications by identifying specific vulnerabilities that arise following the deficiency of a particular gene [55].

Oser MG et al. utilized CRISPR/Cas9 screening methodologies to conduct synthetic lethal screening, aiming to identify therapeutic vulnerabilities in RB1-deficient SCLCs [52]. They discovered that many of the hits encode regulators of chromosomal segregation that functionally interact, including components of condensin complexes (SMC2, NCAPG, and SMC4) and their upstream regulators (AURKB, PLK1, and INCENP). They further explored the intricate interplay between Aurora B inhibition and the transcriptional deregulation of numerous mitotic genes across various RB1-deficient SCLCs. Pharmacologically inhibiting Aurora B by AZD2811 in the SCLC mouse model not only increased polyploidy but also induced cell apoptosis, confirming that AURKB is the Achilles heel of RB1-deficient SCLCs.

Gong X et al. developed a pharmacogenomic screening assay to normalize growth rates by performing two doubling times for each assay and identified that the Aurora A inhibitor LY3295668 showed the strongest cytotoxicity in RB1-mutant SCLCs [56]. They further demonstrated that LY3295668 has the potential to induce cell mitotic defects and cell apoptosis, as evidenced by its efficacy in an in vivo xenograft mouse model with RB1-deficient retinoblastoma [57]. In addition, they highlighted the crucial role of spindle-assembly checkpoint (SAC) activation for the cytotoxicity of LY3295668 in malignancies lacking RB1. On the other hand, an independent study conducted by Lyu J et al., involving an RNAi library in lung cancer cells with RB1 knockout, shed light on the heightened sensitivity of these cells to Aurora A inhibition. The rationale behind this discovery lies in the E2F-mediated elevation of the microtubule destabilizer Stathmin. Inhibition of Aurora A by ENMD-2076 resulted in the activation of Stathmin by decreasing its phosphorylation, disrupting microtubule dynamics, and triggering SAC hyperactivation in RB1-deficient cells, ultimately leading to mitotic cell death [58].

Taken together, these findings underline that the exploitation of synthetic lethality offers promising avenues for addressing the challenges posed by RB1 loss-of-function mutations in tumor suppressor genes. Studies investigating Aurora kinase inhibitors have shown potential in targeting RB1-deficient cancers, suggesting approaches for the development of more effective treatment strategies in oncology.

### 3.3. Synergistic Effect of DNA Damage and PARP Inhibitor on RB1-Deficient Cancers

As previously emphasized, loss-of-function mutations or deletions in tumor-suppressor genes are less favored as targetable driver mutations in cancer therapeutics. Therefore, a classic strategy for treating RB1-deficient malignancies involves the utilization of traditional cytotoxic chemotherapy, which specifically targets rapidly dividing cells. Despite the initial heightened sensitivity of RB1-deficient malignancies to DNA damage, strategies involving combinations of DNA-damaging agents with inhibitors of DNA damage repair, such as poly-ADP ribose polymerase (PARP) inhibitors (PARPis), show promise in addressing RB1-deficient cancers [19,59,60].

R. Vélez-Cruz et al. have elucidated the mechanistic rationale underlying the involvement of RB1 loss in the response to PARPis by revealing a novel, non-transcriptional function for RB1 in homologous recombination (HR) [19]. This finding highlights that RB1 plays a pivotal role in facilitating DNA double-strand break (DSB) repair through HR and underscores the consequences of its absence, which lead to genome instability. In their investigation, it was observed that RB1 localizes to DSBs in a manner dependent on E2F1 and ATM kinase activity. This localization of RB1 is essential for the recruitment of the Brahma-related gene 1 (BRG1) ATPase to DSBs, thus enabling DNA end resection and facilitating the HR process. Depletion of RB1 in breast cancer and osteosarcoma cells results in sensitivity to DNA-damaging drugs, a sensitivity further exacerbated by the PARP inhibitor Olaparib. Supporting their findings, Pietanza MC et al. demonstrated the potential of Temozolomide in combination with the PARP inhibitor Veliparib in treating patients with relapsed-sensitive or refractory SCLCs [59]. Additionally, Zoumpoulidou G demonstrated that RB1-deficient osteosarcoma is selectively sensitive to diverse PARPis such as Olaparib, Niraparib, and Talazoparib [60].

Furthermore, by investigating the impact of RB1 mutation on the transcriptome and proteome, Dong Q et al. revealed that loss-of-function mutations in RB1 increase genomic instability and contribute to the accumulation of double-stranded DNA (dsDNA) within the cytoplasm [61]. Consequently, this accumulation initiates a series of events, including the upregulation of the innate immune signaling response by activating the cGAS/STING signaling pathway, elevated chemokine expression, and triggered immune cell infiltration in lung adenocarcinoma (LUAD). Moreover, xenograft experiments showed that treatment with PARPis reduced tumorigenesis in the A549 RB1-knockout xenograft mouse model. These findings collectively suggest that RB1 mutation mediates sensitivity to PARPis such as Olaparib, Rucaparib, and Niraparib through its crucial role in facilitating DNA DSB repair via HR and in modulating immune responses within the tumor microenvironment.

Another study, conducted by Chakraborty G. et al., investigated the significance of BRCA2 and RB1 co-loss in metastatic castration-resistant prostate cancer (mCRPC) [62]. DNA damage response (DDR) gene variants are frequently observed in both the germline and as somatic abnormalities in mCRPC. Notably, a parallel investigation in prostate cancer patients underscored the significance of BRCA2 germline mutations, particularly when associated with RB1 heterozygous deletion. The co-deletion of BRCA2 and RB1 emerged as an independent genomic driver of castration-resistant prostate cancer (CRPC), contributing to an aggressive phenotype characterized by epithelial-to-mesenchymal transition (EMT). This transition was mediated by the induction of key EMT transcription factors such as SNAIL and SLUG, and the transcriptional co-activator PRRX1. They further highlighted the promising role of PARPis Olaparib and Talazoparib in treating cancers with DDR deficiencies, with a particular focus on the sensitivity of tumors harboring BRCA2 defects. Additionally, the research demonstrated that PARPis Olaparib and Talazoparib significantly hamper the growth of prostate cancer cell lines and organoids derived from human mCRPC exhibiting both homozygous and heterozygous co-deletion of BRCA2 and RB1. Supporting their findings, Miao C. et al. demonstrated that the loss of BRCA2 resensitizes RB1-deleted cells to PARP inhibition in *RNASEH2B*-deleted prostate cancer [63].

Taken together, these findings collectively emphasize the therapeutic potential of targeting DDR pathways, especially in the context of BRCA2 and RB1 co-deletion, to combat the progression of castration-resistant prostate cancer.

### 3.4. Targeting RB1 Loss Cancer with Ferroptosis Inducer

Ferroptosis is a newly recognized, iron-dependent mechanism of programmed cell death, distinguished from conventional cell death pathways like apoptosis, necrosis, or autophagy [64]. Cells die as a result of heightened lipid peroxidation induced by the accumulation of reactive oxygen species (ROS) during ferroptosis, leading to the breakdown of cell membranes and, ultimately, cell death. Long-chain acyl-CoA synthetase 4 (ACSL4) and glutathione peroxidase 4 (GPX4) are two pivotal enzymes, with ACSL4 playing a positive regulatory role and GPX4 exerting a negative influence on the process of ferroptosis.

Growing evidence suggests a connection between ferroptosis and cancer cells that are resistant to therapies or drugs, observed in pancreatic ductal adenocarcinoma, small cell lung cancer (SCLC), triple-negative breast cancer, etc. Wang ME et al. demonstrated that the mechanistic association of ferroptosis links to RB1 tumor suppressor loss and RB1-regulated transcription to targeted therapy [65]. Since E2F is the primary transcription factor (mainly E2F1) driven by RB1 loss to regulate ACSL4 expression in prostate cancer, they unveiled the control of ferroptosis through the RB/E2F/ACSL4 axis, emphasizing the potential therapeutic application of inducing ferroptosis for treating prostate tumor growth driven by RB1 loss. They have reported the significance of RB1 inactivation as a genomic driver of resistance to various targeted therapies, signifying poor clinical outcomes across different cancer types. Additionally, they propose a potential solution by introducing the ferroptosis inducer JKE-1674, a highly selective and stable GPX4 inhibitor that demonstrated selective induction of ferroptosis and lipid peroxidation in prostate cancer cells compared to normal-like prostate epithelial cells, suggesting the feasibility of ferroptosis induction as a promising cancer therapy for RB1-deficient malignancies.

### 3.5. Other Targets

Given the widespread occurrence of loss-of-function mutations in RB1 observed across various human cancers, direct targeting of the RB1 protein presents significant challenges. Nevertheless, emerging evidence points to promising avenues for therapeutic intervention by focusing on downstream targets regulated either through transcriptional modulations or protein–protein interactions by RB1. This shift towards targeting downstream effectors holds the potential for enhancing therapeutic efficacy and achieving more precise treatment outcomes in RB1-mutated cancer patients. Here, we summarize five types of pharmacological inhibition potentially applicable in RB1-deficient cancers (Figure 3).

#### 3.5.1. Targeting RB1 Deficient Tumors through the Ubiquitin-Proteasome Pathway

The intricate regulation of cell cycle progression emphasizes the pivotal role of protein ubiquitination and degradation in facilitating transitions between normal cells and transformed cells. SCF (SKP1-CUL1-Fb-ox) E3 ligase complexes utilize various F-box proteins to recognize and degrade specific substrates [66]. Among these, SKP2, functioning as the substrate recognition F-box protein in SCF^SKP1/CKS1/SKP2^, acts as the E3 ligase that ubiquitinates numerous cell cycle substrates. Consequently, SKP2 plays crucial roles in cell proliferation and differentiation, and is implicated in oncogenesis due to its overexpression in human cancers.

Multiple studies have associated SKP2 with RB1 and demonstrated that SKP2′s activity is regulated by the RB1 tumor suppressor gene [67]. This regulation involves not only mRNA expression [68,69] but also direct protein–protein interaction [70] that modulates SKP2′s enzymatic activity. Loss of RB1 protein function, commonly observed through RB1 mutations in aggressive cancers, leads to dysregulation of SKP2 activity, resulting in decreased levels of p27 and compromised cell cycle control. Given SKP2′s oncogenic nature, inhibiting SKP2 either genetically or pharmacologically holds promise for therapeutic benefits in targeting RB1-deficient tumors.

Upregulation of SKP2 often correlates with the loss of the tumor suppressor RB1, suggesting that SKP2 inhibitors could be effective in tumors driven by various oncogenic drivers. Recognizing that SKP2 loss causes synthetic lethality in RB1 null cells, Aubry and colleagues explored the therapeutic potential of the SKP2 inhibitor MLN4924 in treating retinoblastoma [71]. Their findings revealed that MLN4924 selectively impedes retinoblastoma tumor growth in vitro and in vivo, inducing G1 arrest with apoptosis and G2/M arrest with endoreplication. These results underscore the potential of developing small-molecule SKP2 inhibitors for clinical use in treating RB1-mutant cancers.

#### 3.5.2. Targeting Hyperactive E2F through Histone Demethylase LSD1 Inhibition in RB1-Deficient Tumors

While prior research has shed light on the impact of RB1 inactivation on prostate cancer progression and lineage plasticity, there still remains a critical need to comprehensively understand the underlying mechanisms and identify therapeutic vulnerabilities in RB1-deficient CRPC. Wanting et al. explored the implications of RB1 loss in CRPC, particularly in the neuroendocrine (NE) subtype, given that genomic loss of RB1 is frequently observed in CRPC, correlating with adverse patient outcomes and neuroendocrine transdifferentiation induced by AR signaling inhibition [72]. Intriguingly, the data suggest that the genomic background of prostate cancer cells influences the RB1-E2F transcriptional repression program, with implications for tumor behavior. Through integrated cistromic and transcriptomic analyses, Han W et al. characterized RB1 activity in multiple CRPC models, revealing distinct binding sites and targets based on genomic backgrounds. The study also identified a 49-gene RB1-target signature associated with worse survival in CRPC patients, highlighting the potential of RB1 deficiency as a prognostic marker. Importantly, their findings demonstrated that E2F1 chromatin binding and transcriptional activity in RB1-deficient CRPC are highly reliant on LSD1/KDM1A. Furthermore, RB1 inactivation sensitizes CRPC tumors to LSD1 inhibitor treatment. These findings suggest that LSD1 inhibitors could be effective in treating RB1-deficient CRPC or CRPC-NE, offering a promising avenue for therapeutic development in these aggressive cancer subtypes. This study underscores the importance of further exploring the therapeutic potential of LSD1 inhibitors in treating aggressive forms of prostate cancer characterized by RB1 loss.

#### 3.5.3. Synergistic Chemo-Drug and Histone Methyltransferase DOT1L Inhibition for Treating Retinoblastoma

The dysregulation of epigenetic mechanisms plays a significant role in the tumorigenesis and progression of retinoblastoma [73,74,75]. Despite the identification of aberrantly expressed chromatin regulators in retinoblastoma tumors compared to normal retina, their precise functions in retinoblastoma development and potential as therapeutic targets remain incompletely understood [76]. Mao Y. et al. explored the involvement of the histone H3K79 methyltransferase DOT1L in sensitizing retinoblastoma cells to chemotherapy [77]. Although DOT1L is implicated in promoting leukemia development, and EPZ5676 monotherapy has shown promise in MLL-fusion leukemia, its effectiveness relies on the epigenetic repression of MLL target genes and continuous intravenous infusion to maintain therapeutic concentrations [78]. While DOT1L was found to be expressed in most human retinoblastoma cases, its expression was undetectable in normal retina, suggesting a potential selective vulnerability for DOT1L targeting in retinoblastoma cells. However, treatment with the DOT1L inhibitor EPZ5676 alone demonstrated limited therapeutic efficacy in vitro and in vivo in animal models, primarily due to the need for high doses and sustained drug availability to achieve significant anticancer effects. As DOT1L is involved in DNA damage response and repair, combination treatments of EPZ5676 with genotoxic agents were investigated. Indeed, DOT1L inhibition sensitized retinoblastoma cells to etoposide-induced apoptosis, enhancing the therapeutic effects of DNA-damaging agents. Additionally, the non-histone chromosome protein HMGA2 was identified as a novel target gene of DOT1L in retinoblastoma cells, with its expression epigenetically upregulated by DOT1L. Since HMGA2 promotes retinoblastoma cell proliferation and regulates DNA damage response, these findings suggest that DOT1L inhibition plays a dual role in chemosensitizing retinoblastoma cells by impairing early DNA damage response and downregulating HMGA2 expression.

In the clinical management of retinoblastoma, targeted therapies are not yet established, and conventional chemotherapy remains the standard treatment [79,80]. However, the extensive use of genotoxic drugs in young children with retinoblastoma raises concerns about potential late effects later in life. Targeting epigenetic regulators such as DOT1L may offer a means to enhance the efficacy of current chemotherapy regimens while reducing the doses of genotoxic drugs required for treatment. This approach aligns with recent findings suggesting improved chemotherapy outcomes through combination therapies targeting specific molecular pathways in retinoblastoma xenografts [11]. Therefore, targeting epigenetic dysregulation, particularly through DOT1L inhibition, holds promise for advancing the treatment of retinoblastoma and minimizing the long-term adverse effects associated with conventional chemotherapy.

#### 3.5.4. The Crosstalk of RB1 Loss and ESRRG

It is widely believed that further aberrations are necessary for complete malignant transformation in retinoblastoma [81,82]. However, none of the identified secondary drivers have yet been linked to any clinically targeted therapy. Taking advantage of a large comprehensive multi-omics analysis, Field M.G. et al. conducted a study integrating data from whole-exome sequencing (WES), whole-genome sequencing (WGS), RNA sequencing (RNA-seq), and single-cell RNA-seq (scRNA-seq) to identify previously unknown retinoblastoma dependencies [83]. Their findings revealed that RB1 directly interacts with and inhibits estrogen-related receptor gamma (ESRRG). ESRRG functions as an estrogen-related orphan nuclear receptor transcription factor typically expressed in the developing retina and central nervous system and is known to regulate genes involved in various cellular processes such as development, proliferation, and oxygen metabolism [84].

In the developing retina, a hypoxic environment arises from various factors, including the rapid proliferation of retinal precursor cells, high oxygen demand from newly formed neurons, and restricted blood supply [85,86]. This elevated oxygen demand is further exacerbated by the uncontrolled proliferation of tumor cells lacking RB1 [87,88]. As a result, the absence of RB1 disrupts the negative regulation of ESRRG by RB1, enabling cancer cells to survive under hypoxic stress. Furthermore, pharmacological inhibition by a specific inverse agonist GSK5182 or short hairpin RNA (shRNA)-mediated depletion of ESRRG results in significant RB1-loss cell death, particularly under hypoxic conditions. In clinical observations, heightened expression of ESRRG has been noted in human retinoblastoma tumor cells that infiltrate the optic nerve, a phenomenon strongly correlated with metastatic potential and dismal prognosis. These findings underscore the clinical significance of ESRRG dysregulation in the progression and aggressiveness of retinoblastoma, highlighting its potential utility as a prognostic biomarker and therapeutic target in RB1 loss cancers.

#### 3.5.5. Targeting ER+/RB1-Knockout Breast Cancer with PRMT5 Inhibitor

The approval and clinical utilization of CDK4/6 inhibitors (CDK4/6i) alongside antiestrogen therapy have notably enhanced the progression-free and overall survival rates among patients diagnosed with ER+ metastatic breast cancer [89,90]. Despite these significant advancements, the majority of tumors inevitably develop resistance to this treatment regimen, thereby limiting the available therapeutic options for affected patients. Since RB1 loss-of-function alterations confer resistance to CDK4/6i in ER+ metastatic breast cancer patients [91,92], a genome-wide CRISPR screen identified protein arginine methyltransferase 5 (PRMT5) as a vulnerability in ER+/RB1-deficient breast cancer cells [93]. Mechanistically, inhibition of PRMT5 blocks the G1-to-S transition in the cell cycle independently of RB1 through hyperphosphorylation of Pol II Ser2 and intron retention in multiple genes involved in DNA synthesis, leading to growth arrest in RB1-deficient cells. Additionally, combining the PRMT5 inhibitor pemrametostat with the selective ER degrader fulvestrant synergistically inhibits the growth of ER+/RB-deficient tumors and patient-derived xenografts (PDX) in vivo. These findings suggest dual ER and PRMT5 blockade as a potential therapeutic strategy to overcome CDK4/6i resistance in ER+/RB-deficient breast cancer.

Together, therapeutic targets mentioned above in RB1-deficient cancers along with details on emerging drugs, including their pharmacological function, genome status, and the specific cancer types they target are summarized in Table 1.

## 4. Conclusions and Future Directions

The research summarized above details the multifaceted role of RB1 in cancer biology and its implications for targeted therapy. RB1, a critical tumor suppressor gene, governs various cellular processes, including cell cycle regulation, DNA damage response, and transcriptional modulation. Mutations or dysregulation of RB1 are implicated in the development and progression of diverse cancers, making it a prime target for therapeutic intervention. The canonical function of RB1 in cell cycle control and DNA repair mechanisms underscores its significance in restraining aberrant cell growth and maintaining genomic integrity. However, the complex interplay between RB1 and either lineage-dependent or independent cellular pathways presents challenges in fully elucidating its tumor-suppressive role across different cancer types.

Exploiting vulnerabilities arising from RB1 loss-associated mechanisms offers promising avenues for targeted therapy. Studies investigating alternative RNA splicing dysregulation in RB1-deficient cancers highlight the therapeutic potential of targeting spliceosomal machinery. Furthermore, the identification of Aurora kinase inhibitors and PARP inhibitors as synthetic lethal targets in RB1-deficient malignancies provides insights into the development of more effective treatment strategies. Additionally, targeting RB1 loss-associated cancers with ferroptosis inducers represents a novel therapeutic approach. Ferroptosis induction, particularly through the RB/E2F/ACSL4 axis, emerges as a promising strategy for combating tumor growth driven by RB1 loss, offering potential clinical benefits in various cancer types.

Furthermore, targeting downstream effectors regulated by RB1 presents promising therapeutic approaches for RB1-mutated cancers, overcoming the challenges associated with direct RB1 protein targeting. We highlight nine potential pharmacological strategies in RB1-deficient cancers: inhibition of spliceosomal mechanisms, Aurora Kinase inhibition, PARP inhibition, ferroptosis induction, SKP2 ubiquitin ligase inhibition, histone demethylase LSD1 inhibition, histone methyltransferase DOT1L inhibition, ESRRG inhibition, and arginine methyltransferase PRMT5 inhibition. Each of these inhibitors has been proven effective for different RB1-deficient cancer cells, indicating that targeting downstream effectors regulated by RB1 offers promising strategies for therapeutic intervention in RB1-mutated cancers. This provides insights into potential biomarkers and avenues for precision medicine in cancer treatment.

In addition, numerous potential molecular targets have been identified, but the comprehensive functional validation and prioritization of suitable targets for selective therapy represent the initial steps in developing novel treatments for RB1-mutated cancers. Recent gene expression analyses of human retinoblastoma tissues have unveiled dysregulated chromatin regulators [76], yet their precise influence on retinoblastoma tumorigenesis and progression is still emerging. Analyses of Gene Ontology have revealed that genes related to chromatin/nucleosome assembly and organization, such as DNMT1, DNMT3A DNMT3B, UHRF1, and EZH2, are significantly enriched in human retinoblastoma compared to the normal retina, suggesting these epigenome regulators can be the targets for retinoblastoma and RB1-deficient cancers. With the growing understanding of epigenetic abnormalities in RB tumorigenesis, several elevated chromatin modifiers and associated proteins hold promise for therapeutic interventions.

Given the intricate network of pathways affected by RB1 deficiency and the emerging understanding of downstream effectors, future research could focus on elucidating the crosstalk between RB1 and other key regulatory molecules in cancer cells. Investigating how RB1 loss interacts with other genetic alterations or signaling pathways implicated in cancer development and progression could provide valuable insights into the molecular mechanisms driving RB1-deficient cancers. Additionally, exploring the tumor microenvironment’s role in modulating the effects of RB1 deficiency and identifying potential therapeutic targets within the tumor stroma could open up new avenues for treatment. Integrating multi-omics approaches, including genomics, transcriptomics, proteomics, and metabolomics, could further enhance our understanding of RB1-deficient cancers and facilitate the development of personalized treatment strategies tailored to individual patients’ molecular profiles.

In summary, these findings deepen our understanding of RB1 biology in cancer and provide a foundation for the rational design of targeted therapeutic approaches. By elucidating the molecular mechanisms underlying RB1 deficiency-associated vulnerabilities, these findings pave the way for the development of precision medicine strategies aimed at improving patient outcomes in RB1-deficient malignancies.

## Figures and Tables

**Figure 1 cancers-16-01558-f001:**
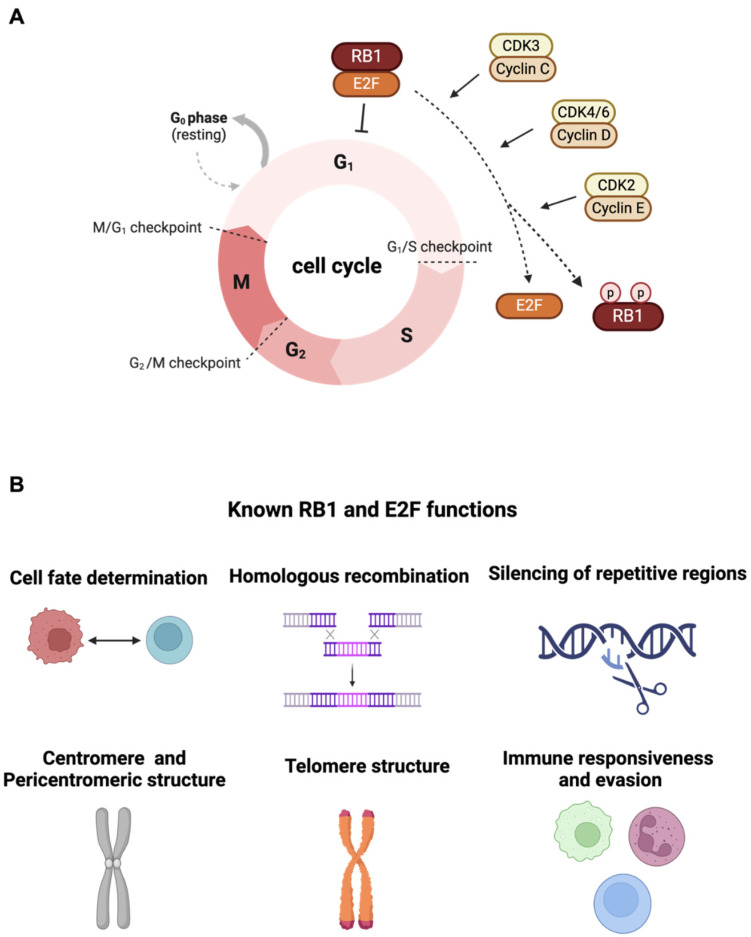
(**A**) The canonical function of RB1 tumor suppressor to arrest cells in the G1 phase of the cell cycle by inhibiting E2F1/2/3 transcription factor family activity is shown. The cell cycle regulators Cyclin C, Cyclin D, and Cyclin E can interact with cyclin-dependent kinases (CDK3, CDK4/6, and CDK2) to induce phosphorylation of RB1 followed by the release of E2F transcription factors. (**B**) Various cellular roles of RB1 include cell fate determination, homologous recombination, silencing of repetitive regions, centromere and pericentromeric structure, telomere structure, and immune responsiveness and evasion (created with BioRender.com).

**Figure 2 cancers-16-01558-f002:**
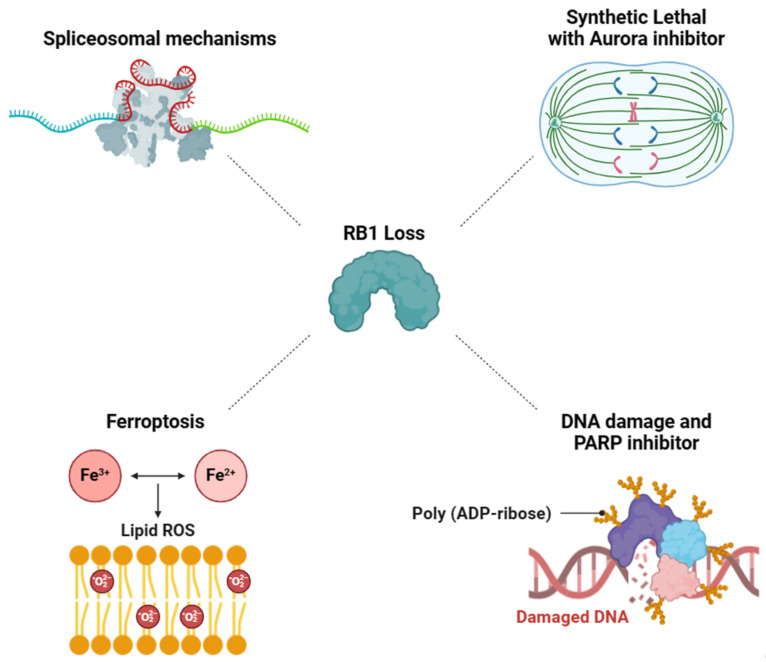
The schematic diagram summarizes four main categories of potential inhibition targets discussed in Section 3.1, Section 3.2, Section 3.3 and Section 3.4. The vulnerabilities associated with RB1 loss are depicted, including (1) Inhibition of spliceosomal mechanisms (**Upper left**), (2) Exploit synthetic lethal with Aurora inhibitor (**Upper right**), (3) Combination therapy to generate DNA damage and PARP inhibitor (**Lower right**), (4) Utilization of a ferroptosis inducer to trigger ferroptosis (**Lower left**) (created with BioRender.com).

**Figure 3 cancers-16-01558-f003:**
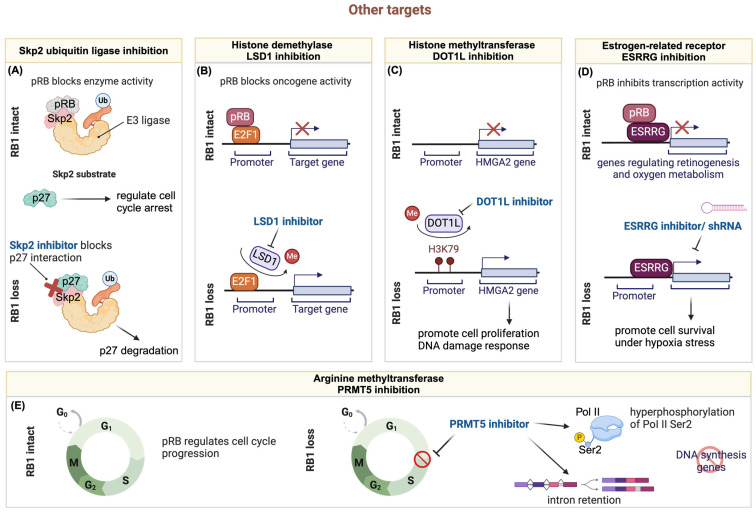
The schematic diagram illustrates the mechanisms of pharmacological inhibition targeting other targets discussed in Section 3.5. (**A**) The schematic depicts the role of RB1 in blocking SKP2 E3 ligase activity by competitively binding to its substrate site. This binding releases p27, thereby regulating cell cycle arrest. In RB1 loss cells, SKP2 binds to and ubiquitinates p27, promoting cell proliferation. SKP2 inhibitors interrupt the interaction between SKP2 and p27. (**B**) RB1 and E2F1 cooperatively bind to the promoters of oncogenes, inhibiting their transcription. In RB1 loss cells, the histone demethylase LSD1 removes methyl groups from the E2F1 transcription factor, activating oncogene transcriptional activity. LSD1 inhibitors suppress the removal of methyl groups from E2F1, downregulating oncogene activity. (**C**) The schematic illustrates the role of the histone methyltransferase DOT1L, which is undetectable in normal retina but highly expressed in RB1 loss cells. High DOT1L levels promote the methylation of the HMGA2 gene promoter, leading to enhanced cell proliferation and DNA damage response. DOT1L inhibition inhibits HMGA2 transcriptional activity and its downstream pathways. (**D**) RB1 inhibits ESRRG transcriptional activity via direct interaction, suppressing genes involved in retinogenesis and oxygen metabolism. In RB1 loss cells, ESRRG activates genes promoting cell survival under hypoxic stress. This activity can be compromised through the utilization of ESRRG inhibitors or shRNA. (**E**) The PRMT5 inhibitor selectively inhibits the growth of RB1-deficient cells by inducing hyperphosphorylation of Pol II Ser2 and intron retention in multiple genes associated with DNA synthesis (created with BioRender.com).

**Table 1 cancers-16-01558-t001:** Summary of pharmacological drugs, their functions, and genome status in relation to RB1-deficient cancers.

Drugs	Pharmacological Function	Cancer Types	Genome Status	References
Pladienolide B	Spliceosomal inhibitors	Osteosarcoma	RB1 mutation	[36]
Sudemycin D6		Osteosarcoma	RB1 mutation	[36]
Barasertib-HQPA (AZD2811)	Aurora kinase B inhibitors	Small Cell Lung Cancer	RB1 deficient	[52]
LY3295668 Erbumine	Aurora kinase A inhibitors	Small Cell Lung Cancer	RB1 deficient	[56]
ENMD-2076		Non Small Cell Lung Cancer	RB1 deficient	[58]
Lynparza (Olaparib)	PARP inhibitors	Lung Adenocarcinoma	RB1 mutation	[61,94]
		Osteosarcoma	RB1 mutation	[60]
Rucaparib		Lung Adenocarcinoma	RB1 mutation	[61]
Zejula (Niraparib)		Lung Adenocarcinoma	RB1 mutation	[61]
		Osteosarcoma	RB1 mutation	[60]
JKE-1674	Ferroptosis inducer	Prostate Cancer	RB1 deficient	[65]
Pevonedistat (MLN4924)	SKP2 inhibitors	Retinoblastoma	RB1 deficient, MYC amplification	[71]
GSK2879552	LSD1 inhibitors	Prostate Cancer	RB1 deficient	[72]
GSK5182	ESRRG inhibitors	Retinoblastoma	RB1 deficient	[83]
Pemrametostat (GSK3326595)	PRMT5 inhibitors	Breast Cancer	ER+, RB1 deficient	[93]

## Data Availability

The data presented in this article are available in the references provided.

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
