# Peer review of "Therapeutic Strategies for RB1-Deficient Cancers: Intersecting Gene Regulation and Targeted Therapy"

_cancers, 2024, doi:10.3390/cancers16081558_

Round 1

Reviewer 1 Report

Comments and Suggestions for Authors

The review by Huang et al. focuses on cellular pathways that are potential targets of therapy in RB1-deficient cancers. Authors discuss both canonical and non-canonical functions of RB1 that are involved in its tumor suppression mechanism, including discussions on the RB1/E2F-pathways, spliceosomal mechanism, DNA damage, Mitosis, and other mechanisms. Authors also discussed drugs that exhibited effects on RB1-negative cancers. However, the authors did not bring up discussions on RB1 role in differentiation or its interaction with FoxM1, a critical transcription factor in cancer.

Addressing the following concerns will improve the review.

1.     The review left out the roles of RB1 in differentiation – likely a significant mechanism in tumor suppression. 

2.     Authors did not discuss the interaction of RB1 with FoxM1 – That is a significant omission because FoxM1 is a driver for aggressive cancer progression. 

3.     Inclusion of a table with the inhibitors (or drugs), RB1 status, and cancer types would be useful.

Author Response

Point-by-point response to reviewers’ and editorial comments

We thank all the reviewers for their positive assessment of our work and their highly constructive comments. This has helped us to revise the manuscript to greatly improve the clarity and the impact of our findings on the field.  For the reviewers’ convenience, we have made edits in the revised manuscript and highlighted the major changes.

Reviewer #1

Comments and Suggestions for Authors

The review by Huang et al. focuses on cellular pathways that are potential targets of therapy in RB1-deficient cancers. Authors discuss both canonical and non-canonical functions of RB1 that are involved in its tumor suppression mechanism, including discussions on the RB1/E2F-pathways, spliceosomal mechanism, DNA damage, Mitosis, and other mechanisms. Authors also discussed drugs that exhibited effects on RB1-negative cancers. However, the authors did not bring up discussions on RB1 role in differentiation or its interaction with FoxM1, a critical transcription factor in cancer.

Addressing the following concerns will improve the review.

  1. The review left out the roles of RB1 in differentiation – likely a significant mechanism in tumor suppression.

Response: We appreciate the reviewer's suggestion, and as a result, we have incorporated information regarding RB1's role in cell differentiation into our introduction (line 95-96, line107-130).

  1. Authors did not discuss the interaction of RB1 with FoxM1 – That is a significant omission because FoxM1 is a driver for aggressive cancer progression.

Response: Based on the reviewer’s suggestion, we have included a discussion of the interaction of RB1 and FOXM1 in our review article (line 132-150).

  1. Inclusion of a table with the inhibitors (or drugs), RB1 status, and cancer types would be useful.

Response: Based on the reviewer’s suggestion, we have included a table in our review (line 505-511, Table 1)

Reviewer 2 Report

Comments and Suggestions for Authors

Rb1 is one of the most lost genes in several cancers. Considering its importance in Cancer, not surprisingly Rb1 is one of the most studied tumor suppressor genes. There are numerous review articles has been published in last 2-3 years on the different therapeutic strategies for Rb1-deficient cancers and the present manuscript failed to add anything new on this subject.

Author Response

Point-by-point response to reviewers’ and editorial comments

We thank all the reviewers for their positive assessment of our work and their highly constructive comments. This has helped us to revise the manuscript to greatly improve the clarity and the impact of our findings on the field.  For the reviewers’ convenience, we have made edits in the revised manuscript and highlighted the major changes.

Reviewer #2

Comments and Suggestions for Authors

Rb1 is one of the most lost genes in several cancers. Considering its importance in Cancer, not surprisingly Rb1 is one of the most studied tumor suppressor genes. There are numerous review articles has been published in last 2-3 years on the different therapeutic strategies for Rb1-deficient cancers and the present manuscript failed to add anything new on this subject.

Response: We appreciate the reviewer's feedback regarding the timeliness of certain articles. Given that our review primarily centers on therapeutics targeting RB1-deficient cancers, we have made every effort to encompass all pertinent research findings, not confining ourselves solely to the past 1-2 years. Upon revision, we've identified a new article pertinent to these topics and have consequently incorporated it into our review (line 366-368, line 381-383, line 488-503, line 549-560).

Reviewer 3 Report

Comments and Suggestions for Authors

The authors provide a well-constructed and comprehensive review of the role of RB1 in cancer biology and the implications of these roles for targeted therapy. They try to clarify the complex interplay of RB1 and the various cellular pathways involved in cancer progression. They also highlight the therapeutic potential due to vulnerabilities arising from RB1 deficiencies, significantly contributing to our understanding of tumour biology and therapeutic strategies.

Specifically,

  1. Reviewing RB1's canonical function and the significant associated cellular processes, such as cell cycle regulation and DNA repair, strengthens our understanding of its multifaceted role in cancer biology and provides a solid foundation for further perspectives.
  2. The potential therapeutic strategies exploiting RB1 loss-associated vulnerabilities across diverse cancer types are outlined. This can be a valuable guide for future research directions.
  3. The authors also provided a comprehensive summary of targeting downstream effectors regulated by RB1 as a promising therapeutic approach, an aspect rarely discussed in depth in previous literature.
  4. The discussion on the intricacies of multi-comics approaches could potentially facilitate the development of "tailor-made" strategies for RB1-deficient cancers.

Overall, this manuscript provides a well-rounded perspective on RB1 in cancer biology. The summary of complex interactions and the review of potential targeted therapies will serve as a valuable resource for researchers in the field. It provides ideas for future studies that elucidate crosstalk between RB1 and other critical regulatory molecules in cancer cells.

Author Response

Point-by-point response to reviewers’ and editorial comments

We thank all the reviewers for their positive assessment of our work and their highly constructive comments. This has helped us to revise the manuscript to greatly improve the clarity and the impact of our findings on the field.  For the reviewers’ convenience, we have made edits in the revised manuscript and highlighted the major changes.

Reviewer #3

Comments and Suggestions for Authors

The authors provide a well-constructed and comprehensive review of the role of RB1 in cancer biology and the implications of these roles for targeted therapy. They try to clarify the complex interplay of RB1 and the various cellular pathways involved in cancer progression. They also highlight the therapeutic potential due to vulnerabilities arising from RB1 deficiencies, significantly contributing to our understanding of tumour biology and therapeutic strategies.

Specifically,

  1. Reviewing RB1's canonical function and the significant associated cellular processes, such as cell cycle regulation and DNA repair, strengthens our understanding of its multifaceted role in cancer biology and provides a solid foundation for further perspectives.
  2. The potential therapeutic strategies exploiting RB1 loss-associated vulnerabilities across diverse cancer types are outlined. This can be a valuable guide for future research directions.
  3. The authors also provided a comprehensive summary of targeting downstream effectors regulated by RB1 as a promising therapeutic approach, an aspect rarely discussed in depth in previous literature.
  4. The discussion on the intricacies of multi-comics approaches could potentially facilitate the development of "tailor-made" strategies for RB1-deficient cancers.

Overall, this manuscript provides a well-rounded perspective on RB1 in cancer biology. The summary of complex interactions and the review of potential targeted therapies will serve as a valuable resource for researchers in the field. It provides ideas for future studies that elucidate crosstalk between RB1 and other critical regulatory molecules in cancer cells.

Response: Thanks to the reviewer for providing a summary of our review, and we truly value your acknowledgment of its significance as a valuable resource for research in the field.

Round 2

Reviewer 1 Report

Comments and Suggestions for Authors

No more comments.

Reviewer 2 Report

Comments and Suggestions for Authors

Authors revised the manuscript as per suggestions.